# Clinical Potential of Circulating Cell-Free DNA (cfDNA) for Longitudinally Monitoring Clinical Outcomes in the First-Line Setting of Non-Small-Cell Lung Cancer (NSCLC): A Real-World Prospective Study

**DOI:** 10.3390/cancers14236013

**Published:** 2022-12-06

**Authors:** Valerio Gristina, Nadia Barraco, Maria La Mantia, Luisa Castellana, Lavinia Insalaco, Marco Bono, Alessandro Perez, Delia Sardo, Sara Inguglia, Federica Iacono, Sofia Cutaia, Tancredi Didier Bazan Russo, Edoardo Francini, Lorena Incorvaia, Giuseppe Badalamenti, Antonio Russo, Antonio Galvano, Viviana Bazan

**Affiliations:** 1Department of Surgical, Oncological and Oral Sciences, University of Palermo, 90127 Palermo, Italy; 2Department of Experimental and Clinical Medicine, University of Florence, 50134 Florence, Italy; 3Department of Experimental Biomedicine and Clinical Neurosciences, School of Medicine, University of Palermo, 90127 Palermo, Italy

**Keywords:** NSCLC, liquid biopsy, cfDNA, treatment monitoring, ECOG-PS 2

## Abstract

**Simple Summary:**

In the precision oncology era, liquid biopsy has dramatically revolutionized the management of such patients, potentially overcoming tissue biopsy limitations while entering the current clinical practice as a valuable diagnostic tool. However, despite the increasing implementation of targeted and immunotherapy-based treatments, the prognosis of patients with advanced non-small-cell lung cancer (NSCLC) remains dismal. We prospectively evaluated longitudinal plasma cfDNA kinetics as an early marker of therapeutic efficacy in patients with advanced NSCLC undergoing standard first-line treatments. Our real-world study demonstrates that quantitative changes in cfDNA values correlate with responses to therapy and relapse of the disease in treatment-naïve patients with advanced NSCLC undergoing TKI- and IO-based treatments.

**Abstract:**

Background: Despite the increasing implementation of targeted and immunotherapy-based treatments, the prognosis of patients with advanced NSCLC remains dismal. We prospectively evaluated longitudinal plasma cfDNA kinetics as an early marker of therapeutic efficacy in patients with advanced NSCLC undergoing standard first-line treatments. Methods: From February 2020 to May 2022, treatment-naïve patients with advanced NSCLC were consecutively enrolled at the Medical Oncology Unit of the Paolo Giaccone University Hospital, Palermo (Italy). We quantified cfDNA in terms of ng/μL using a Qubit^TM^ dsDNA HS Assay Kit. The agreement between the cfDNA and radiologic response was evaluated from baseline (T0) to the radiologic evaluation (T1). Results: A total of 315 liquid biopsy samples were collected from 63 patients at baseline, with a total of 235 paired plasma samples from 47 patients at disease re-evaluation. A fair concordance was observed between early and durable radiographic and cfDNA response (Cohen’s kappa coefficient = 0.001); 11 and 18 patients receiving TKI (Pearson’s chi-squared test = 4.278; Cohen’s kappa coefficient = 0.039) and IO treatments (Pearson’s chi-squared test = 7.481; Cohen’s kappa coefficient = 0.006) showed a significant and durable association between cfDNA dynamics and the first radiologic evaluation, whereas among the 18 patients undergoing CT, no significant correlation was observed (Pearson’s chi-squared test = 0.720; Cohen’s kappa coefficient = 0.396). The ECOG-PS 2 patients presented with the mean baseline cfDNA levels 2.6-fold higher than those with ECOG-PS 0–1 (1.71 vs. 0.65 ng/µL; *p* = 0.105). Conclusions: Our real-world study demonstrates that quantitative changes in cfDNA values correlated with responses to therapy and relapse of disease in treatment-naïve patients with advanced NSCLC undergoing TKI- and IO-based treatments.

## 1. Introduction

Despite the increasing implementation of targeted and immunotherapy-based treatments, the prognosis of patients with advanced non-small-cell lung cancer (NSCLC) remains dismal [1]. In the precision oncology era, liquid biopsy has dramatically revolutionized the management of such patients, potentially overcoming tissue biopsy limitations while entering the current clinical practice as a valuable diagnostic tool [2,3]. In this regard, besides the clinical utility for predictive purposes in oncogene-addicted NSCLC, the adoption of liquid biopsy testing may represent a useful tool for monitoring clinical outcomes and prognostication [4,5]. Namely, the quantification of circulating tumor DNA (ctDNA), the component of cell-free DNA (cfDNA) released from tumor sites into the bloodstream of cancer patients, emerged as a minimally invasive approach to real-time monitoring of the tumor evolution using ctDNA kinetics as a potential efficacy predictor for patients with advanced NSCLC [6,7]. However, despite the growing body of evidence in the literature, the quantitative monitoring of cfDNA for predicting radiological responses to standard treatments in NSCLC has not yet entered the clinical practice [8,9]. The peculiar advantages of using dynamic in vivo biomarkers make cfDNA an appealing tool for therapeutic monitoring during anticancer therapies, mostly considering that different meta-analyses have already proved with a high level of overall accuracy that the amount of cfDNA is higher in patients with lung cancer than in healthy individuals [10,11]. Hence, additional data validating the role of cfDNA in predicting and monitoring clinical outcomes in the first-line setting of NSCLC are warranted.

In this real-world study, we prospectively evaluated longitudinal plasma samples to investigate the potential of cfDNA kinetics as an early marker of therapeutic efficacy and predictor of prolonged survival in patients with NSCLC undergoing standard first-line treatments.

## 2. Materials and Methods

### 2.1. Patients and Study Design

This is a real-world prospective cohort study including the systematic assessment of tumor tissue biopsies at baseline and the monitoring of treatment-induced changes in the blood profile in treatment-naïve advanced NSCLC patients who were candidates for standard first-line treatments based on the predictive molecular pathology and clinicopathological characteristics. From February 2020 to May 2022, patients with advanced NSCLC were consecutively enrolled at the Medical Oncology Unit of the Paolo Giaccone University Hospital, Palermo (Italy). Tumor tissues were obtained via systematic baseline biopsies and stored as formalin-fixed paraffin-embedded (FFPE) samples at Department “G. D’Alessandro”, Pathology Institute, University of Palermo, and at other referring pathology units. Paired whole blood samples were collected at baseline and the first radiologic evaluation of disease (within 12 weeks) according to a standardized protocol and stored frozen. All the FFPE and plasma samples were analyzed at the Laboratory of Molecular Oncology at the Regional Reference Center for the Prevention, Diagnosis, and Treatment of Rare Heredofamilial Cancers of Adult Medical Oncology (Medical Oncology Unit, Department of Surgical, Oncological, and Oral Sciences, A.O.U.P. “P. Giaccone”, University Hospital of Palermo), an accredited Italian reference genetic center for prognostic and predictive molecular testing in oncology. FFPE tissue collection, nucleic acids extraction, and molecular analysis are comprehensively described in the Appendix A.

All the patients underwent a CT scan every 3 months, and the radiologic responses were classified according to the response evaluation criteria in solid tumors (RECIST), version 1.1. The CT scans were collected at baseline and every 3–6 months as per clinical practice [12]. The agreement between the cfDNA response and the radiographic tumor response was evaluated in patients with available plasma samples from baseline (T0) to the radiologic evaluation (T1) and at least two clinical follow-ups for therapeutic assessment. Clinical and pathological characteristics of all the recruited patients including demographics, baseline clinical features, tumor- and treatment-related data were retrieved from the available clinical records. The inclusion criteria were as follows: (1) Eastern Cooperative Oncology Group (ECOG) performance status (PS) ≤ 2; (2) patients with histologically or cytologically documented NSCLC with unresectable stage IIIB–C or Stage IV disease (according to the 8th edition of IASLC TNM) who were treatment-naive and eligible for the first-line tyrosine kinase inhibitors (TKIs) (osimertinib, alectinib, crizotinib, or the dabrafenib + trametinib combination), immuno-oncology (IO)-based treatment (pembrolizumab ± platinum-based chemotherapy (CT)), or CT only. The exclusion criteria were as follows: (1) patients with other malignant tumors; (2) patients with ECOG-PS ≥ 3; (3) patients who received prior first-line TKIs or IO-based treatment ± platinum-based chemotherapy; (4) patients with a mental illness preventing the signing of the informed consent form. The study was approved by the local ethics committee according to the principles outlined in the Declaration of Helsinki. The study was conducted in accordance with the Declaration of Helsinki, and the protocol was approved by Ethics Committee Palermo I (AIFA code CE 150109, Statement No. 02/2020, approved on 19 February 2020). Written informed consent was obtained from all the enrolled patients.

### 2.2. Plasma Separation, DNA Extraction, cfDNA Quantification, and Molecular Analysis

Blood samples (~5 mL) were collected into K2 EDTA tubes at baseline prior to the first drug administration and at each instrumental disease re-evaluation during the treatment course. They were immediately processed for plasma collection and centrifuged twice (10 min at 3000 rpm; 10 min at 16,000× *g*). Sample processing was carried out within 2 h from plasma collection. The collected plasma samples were used to extract cfDNA. We extracted cfDNA from 1 to 2 mL of plasma using a QIAamp Circulating Nucleic Acid Kit (Qiagen) and quantified it in terms of ng/μL using a Qubit^TM^ dsDNA HS Assay Kit. Cell-free nucleic acids (cfNAs) were analyzed in dynamics using an Oncomine^TM^ Lung cfTNA Research Assay. Every single next-generation sequencing (NGS) run datum was compared with a positive in-house control as a validation set. The libraries were quantified using an Ion Library TaqMan^TM^ quantification kit on a QuantStudio7 Pro Real-Time PCR System (Applied Biosystems) using Design and Analysis Software v2.4.3. Using 20 ng of cfNAs, the specificity of this kit was 99% at 0.1% of the limit of detection (LOD). The data were tested on an amplicon-based sequencing platform Ion Torrent S5^TM^ System. Oncomine TagSeq Lung v2 Liquid Biopsy-w2.5-Single Sample was the workflow applied for the analysis of cfNAs samples. To test the reliability of the data for cfNA sequencing, we used the following thresholds: total mapped reads > 3M, median read coverage Avg 40,000–Min > 25,000, median molecular coverage > 2500. The data of DNA sequencing were analyzed with Ion Torrent TorrentSuite^TM^ (TS, version 5.18) using the Coverage Analysis and Variant Caller plugins. The sequencing data were categorized by relevance with the related percentage of allelic frequency as annotated by Ion Reporter Software v5.18 applying the Variant Matrix Summary (5.18) filter chains for default use.

### 2.3. Statistical Analysis

Descriptive statistics were used to analyze demographic and clinicopathological data. According to radiologic response, the disease was defined as responsive (complete response (CR) or partial response (PR)) or non-responsive (stable disease (SD) or progressive disease (PD)) based on the standard RECIST 1.1. Regarding the cfDNA levels, we dichotomized values as ≥ or <20% indicating the change from the baseline cfDNA to higher and lower levels, respectively, after the beginning of treatment. We used X-tile analysis to determine the optimal cfDNA cutoff value for survival prediction according to progression-free survival (PFS) and overall survival (OS) [13]. A paired Wilcoxon test was used to compare the median cfDNA plasma levels before and after the beginning of treatment. Furthermore, the mean differences between the clinically relevant subgroups of interest (sex, smoking habits, and age ≥65) were explored using the nonparametric Mann–Whitney U test. The Pearson correlation coefficient between the baseline cfDNA and the clinical parameters was explored. Cohen’s kappa test was used to determine the concordance of dynamic changes in the liquid biopsy data and radiologic response with a 95% confidence interval (CI). Pearson’s chi-squared test or Fisher’s exact test was used to determine any statistically significant association between cfDNA dynamics and the radiologic response to systemic treatments. The Kaplan–Meier method was used for performing survival analysis, providing the median and *p*-values, using the logrank test for comparisons. Univariate and multivariate analyses were performed using the Cox proportional hazards and logistic regression models. The multivariable model included as the covariates all the pretreatment parameters found to have a *p*-value < 0.05 in univariate analysis. A *p*-value < 0.05 was used as the threshold for statistical significance. All the statistical analyses were performed using version 20 of the SPSS statistics software (IBM, Armonk, NY, USA).

## 3. Results

### 3.1. Clinicopathological Characteristics

Of the 87 patients who met the inclusion criteria, 73 had complete clinical records and were prospectively enrolled. The clinicopathological characteristics of the study population are shown in Table 1. Most patients were ≥65 years old (54.8%), male (71.2%), current or former smokers (76.7%), presenting with adenocarcinoma histology (76.7%) and ECOG-PS 0–1 (57.5%). Notably, a significant portion (42.5%) of the included patients had ECOG-PS 2. The majority presented with locoregional lymph node involvement (84.9%), with the most common distant metastatic sites represented by bone (32.8%), central nervous system (CNS; 19.2%), adrenal gland (17.8%), and liver (12.3%).

### 3.2. Molecular Diagnostics

Table 2 comprehensively summarizes the molecular diagnostics using either tissue or blood. Regarding tissue diagnostics, 55 (75.3%) non-squamous samples were assessed for *EGFR* or *BRAF* mutations using RT-PCR with the total mutation rate of 20% (11/55), detecting eight and three hotspot point mutations, respectively. On the other hand, even if only nine (12.3%) tissue specimens were evaluated, the total mutation rate using DNA/RNA-based NGS was 66% (6/9), covering nine different activating genomic alterations within the *EGFR*, *KRAS*, *ALK*, *MET*, *RET*, and *ROS1* genes. Finally, five (6.8%) patients underwent plasma genotyping via NGS.

### 3.3. Clinical Outcomes

Treatment based on TKIs, IO, and CT was received by 19, 28, and 26 patients, respectively. At the time of survival analysis (median follow-up of 20.7 months, range: 17.3–24.1 months), 51 patients had disease progression, while 42 patients died because of tumor progression, with 31 patients still being alive at the time of data analysis. In the overall population, the median PFS and OS were 6.1 (95% CI: 4.0–8.2) and 12.8 (95% CI: 2.1–23.5) months, respectively. Among the specific treatment subgroups, the median PFS and OS were 6.0 (95% CI: 0–25.5) and 32.6 (95% CI: 0–72.7) months, 10.3 (95% CI: 0–24.3) and 20.5 (95% CI: 6.9–34.1) months, and 4.2 (95% CI: 2.5–5.9) and 9.0 (95% CI: 6.3–11.6) months in the patients receiving TKIs, IO-based treatment, and CT, respectively (Appendix A). Of note, when compared to CT only, the patients receiving TKI- or IO-based treatments had significantly improved PFS (*p* = 0.022 and *p* = 0.006, respectively) and OS (*p* = 0.054 and *p* = 0.057, respectively).

### 3.4. Prognostic Value of the Baseline cfDNA Levels

Briefly, a total of 315 liquid biopsy samples were collected from 63 patients at baseline, with a total of 235 paired plasma samples from 47 patients at disease re-evaluation. Among the 63 patients evaluable for cfDNA analysis at baseline, the median cfDNA level was 0.61 ng/µL, thus not significantly higher than that observed in the 47 patients evaluated at the first follow-up point (0.57 ng/µL, Wilcoxon rank-sum test *p* = 0.536) (Figure 1 and Figure 2). Of note, considering those clinical parameters potentially modified the baseline cfDNA values, no statistically significant differences were reported for gender (Mann–Whitney U test *p* = 0.610), age (*p* = 0.476), and smoking (*p* = 0.183).

Categorizing the overall population by the median cfDNA value (0.61 ng/µL) into low and high groups, significant differences in the median PFS (8.4 months, 95% CI: 2.5–14.3 months vs. 4.2 months, 95% CI: 2.5–5.9 months; *p* = 0.043) and OS (30.3 months, 95% CI: 18.4–42.1 months vs. 4.7 months, 95% CI: 2.6–6.9 months; *p* < 0.0001) for the patients with lower vs. higher cfDNA levels, respectively, were observed (Appendix A).

Predictably, no significant correlation was shown between the baseline cfDNA and tumor size (mean, 5 cm) in our population (R = −0.078; *p* = 0.543), probably due to the extra-thoracic burden of the disease contributing to cfDNA shedding.

To enhance the prognostic accuracy of the baseline cfDNA levels, the patients were further dichotomized into low and high cfDNA groups according to a refined threshold based on the X-tile analysis. In the all-comers population, a baseline cfDNA cutoff value of 0.68 ng/µL seemed to reliably discriminate both for PFS and OS between the patients with good and poor prognosis (Appendix A). Consistently, significant differences in the median PFS (8.3 months, 95% CI: 3.3–13.4 months vs. 4.5 months, 95% CI: 3.2–5.8 months, *p* = 0.038) and OS (23.3 months, 95% CI: 9.7–36.9 months vs. 4.5 months, 95% CI: 3.4–5.5 months; *p* < 0.0001) were found in the two cfDNA categories (low vs. high baseline levels, respectively; *p* < 0.0001).

In the oncogene-addicted subgroup, we observed a baseline cfDNA cutoff value of 0.92 ng/µL for the PFS (median PFS = 24.0 months, 95% CI: 0–48.6 months versus median PFS = 2.5 months, 95% CI: 0–5.1 months in the low and high cfDNA groups, respectively), even if not statistically significant (*p* = 0.293); on the other hand, the patients receiving TKIs with cfDNA concentrations higher than 0.68 ng/µL had a significantly shorter OS (median OS = 4.0 months, 95% CI: 2.9–5.0 months) than those with lower cfDNA levels (median OS = 32.6 months, 95% CI: 0–76.5 months) (*p* = 0.044) (Appendix A). The patients treated with IO-based regimens and having baseline cfDNA levels higher than 0.65 ng/µL experienced poorer PFS (median PFS = 6.1 months, 95% CI: 0–14.0 months) and OS (median OS = 6.1 months, 95% CI: 0.1–12.0 months) when compared to the patients with lower cfDNA concentrations (median PFS and OS = not reached (NR)) (*p* = 0.021 and 0.012, respectively) (Appendix A). Those patients undergoing CT with the baseline cfDNA levels higher than 0.63 ng/µL and 0.50 ng/µL had significantly shorter PFS (median PFS = 3.9 months; 95% CI: 0.4–7.3 months) and OS (median OS = 5.7 months; 95% CI: 3.5–7.9 months) than those with lower cfDNA concentrations (median PFS = 6.8 months; 95% CI: 6.1–7.4 months; median OS = 20.2 months; 95% CI: 14.7–25.7 months) (*p* = 0.022 and 0.018, respectively) (Appendix A).

### 3.5. Dynamic Plasma cfDNA Values Are Associated with Radiologic Response and Survival

We compared the baseline and post-treatment cfDNA levels between the responders (complete or partial response, N = 23) and non-responders (stable or progressive disease, N = 24). A 20% cfDNA increase from baseline was detected as the median increase and used as the cutoff point for survival analysis. 

While 16/20 (80%) patients presenting with at least a 20% increase did not experience a disease response at first restaging, 19/27 (70.4%) subjects with a sharp drop in the cfDNA level showed a prompt response to systemic treatments (Pearson’s chi-squared test = 11.665). Interestingly, when assessing the agreement between the radiographic and cfDNA response from T0 to T1, a fair concordance for a 20% cfDNA response was observed between the early and durable radiographic and cfDNA responses (Cohen’s kappa coefficient = 0.001). No significant differences were reported between the radiologic and liquid biopsy response assessments (median value, 8 days), reducing the risk of misleading results.

Dealing with the treatment subgroups, 11 and 18 patients receiving TKIs (Pearson’s chi-squared test = 4.278; Cohen’s kappa coefficient = 0.039) and IO-based treatment (Pearson’s chi-squared test = 7.481; Cohen’s kappa coefficient = 0.006) showed a significant and durable association between cfDNA dynamics and the first radiologic evaluation, whereas among the 18 patients undergoing CT, no significant correlation was observed (Pearson’s chi-squared test = 0.720; Cohen’s kappa coefficient = 0.396).

We strived to associate cfDNA dynamics with survival outcomes to provide further clinical insights. Overall, within the cfDNA responsive group, 27/47 (57%) patients had a significantly improved median PFS (18.9 months; 95% CI: 6.2–31.5) when compared to 20/47 (43%) cfDNA non-responders (3.3 months; 95% CI: 2.9–3.8) (*p* = 0.004) (Figure 3). Conversely, no benefit in terms of the OS was observed (30.3 months, 95% CI: 12.2–48.3 vs. 20.5 months, 95% CI: 14.4–26.6, respectively; *p* = 0.133) (Figure 3). Considering the PFS results, we stratified the survival data according to the treatment subgroups. Compared to the cfDNA non-responders, the cfDNA responsive patients receiving TKIs and IO-based treatment experienced a numerically longer PFS (24.0 months (95% CI: 0–52.3) vs. 2.5 months (95% CI: 0–18.3) and NR vs. 3.4 months (95% CI: 0–19.6), respectively), although not formally statistically significant (*p* = 0.219 and 0.338, respectively). On the other hand, the cfDNA responders undergoing CT showed a significantly improved survival in terms of both clinical and statistical relevance (7.6 months (95% CI: 5.0–10.2) vs. 3.2 months (95% CI: 1.5–4.9), *p* = 0.025).

### 3.6. Survival Outcomes and Multivariate Analysis

Multivariate Cox proportional regression analyses were performed to assess whether a cfDNA increase over the first 12 weeks of therapy represented an independent factor related to the effectiveness of systemic treatments in terms of the PFS and OS.

The multivariate analyses identified the presence of liver metastases (hazard ratio (HR), 0.027; 95% confidence interval (CI), 0.004–0.175; *p* < 0.0001) and a cfDNA increase > 20% (HR, 0.345; 95% CI, 0.165–0.722; *p* = 0.005) as factors significantly associated with worse PFS (Appendix A). Interestingly, regarding the OS, multivariate analyses confirmed the occurrence of liver metastases (HR, 0.314, 95% CI, 0.14–0.697, *p* = 0.004) as a variable associated with worse survival while further revealing ECOG-PS 0–1 and a lower median cfDNA as independent prognostic factors for the OS. Accordingly, ECOG-PS 0 was associated with a significantly reduced risk of death (HR, 0.22, 95% CI, 0.08–0.614, *p* = 0.004) compared with ECOG-PS 2 (Appendix A).

### 3.7. The Predictive Role of ECOG-PS

The role of ECOG-PS in the overall cohort population was further explored, even according to the available matched cfDNA samples. Overall, compared to the patients with ECOG-PS 0 or 1, the ECOG-PS 2 patients seemed to experience significantly poorer median PFS (4.2 (95% CI, 2.3–6.1) vs. 8.3 (95% CI, 3.5–13.1) months; *p* = 0.024) and OS (6.1 (95% CI, 2.5–9.7) vs. 23.3 (95% CI, 12.0–34.6) months; *p* < 0.0001).

Across the treatment subgroups, the median PFS was numerically lower in the ECOG-PS 2 patients receiving TKIs, even if not formally reaching statistical significance, compared to the ECOG-PS 0–1 subjects (4.0 (95% CI, 0–8.3) months vs. 24.0 (95% CI, 0–57.4); *p* = 0.123). Similarly, the patients with a poorer PS receiving IO-based treatments had a numerically shorter median PFS while showing a significant trend for statistical significance (6.1 (95% CI, 0.6–11.5) months vs. NR; *p* = 0.088). On the contrary, no statistically significant differences between the ECOG-PS 2 and 0–1 patients undergoing only CT regimens were observed in terms of the PFS (3.2 (95% CI, 2.1–4.3) months vs. 6.5 (95% CI, 1.5–11.4); *p* = 0.354).

Regarding the OS, the ECOG-PS 2 patients receiving TKIs or CT had clinically and statistically a poorer survival (4.0 (95% CI, 3.2–4.8) months vs. 32.6 (95% CI, NR–NR), *p* = 0.003; 4.8 (95% CI, 0–9.9) months vs. 18.5 (95% CI, 1.9–35.0), *p* = 0.039, respectively) than those with ECOG-PS 0–1. Likewise, even if only showing a strong trend for formal statistical significance, the ECOG-PS 2 patients undergoing IO-based treatments exhibited a poorer survival (12.1 (95% CI, 3.9–20.4) months vs. NR, *p* = 0.074). Intriguingly, when comparing individual patients receiving pembrolizumab in association (15/28, 53.5%) or not (13/28, 46.5%) with chemotherapy within the IO-based subgroup, the ECOG-PS 2 patients receiving the combination approach seemed to experience a poorer survival in terms of both the PFS (*p* = 0.015) and OS (*p* = 0.036) as compared to single-agent pembrolizumab (*p* = 0.842 and *p* = 0.644 for the PFS and OS, respectively), suggesting a possible predictive value of ECOG-PS in patients on IO combinations.

Finally, among the 26/63 (41.2%) and 37/63 (58.8%) patients presenting with ECOG-PS 2 and 0–1 evaluable for cfDNA kinetics, neither significant association nor correlation between the early cfDNA and the radiologic response was observed (Pearson’s chi-squared test = 0.003; Cohen’s kappa coefficient = 0.959). Remarkably, even if not formally significant, ECOG-PS 2 patients presented with the mean baseline cfDNA levels 2.6-fold higher than those with ECOG-PS 0–1 (1.71 vs. 0.65 ng/µL; *p* = 0.105).

## 4. Discussion

This prospective biomarker trial confirmed the independent prognostic value of baseline cfDNA [14,15,16] and demonstrated the durable association of cfDNA dynamics with treatment response in the real-life setting of patients with advanced NSCLC receiving first-line TKI- and IO-based treatments. In contrast, consistently with other previously published reports [17,18], cfDNA kinetics did not appear to fairly predict the CT response while showing comparable quantitative levels according to the disease burden that seemed to be in line with the literature [19,20,21,22].

Even if emerging as a useful method for real-time monitoring of the efficacy of targeted therapies, the evaluation of cfDNA for the efficacy of CT or IO combinations has been controversial [23,24,25,26]. Several research groups have recently suggested the increasing role of cfDNA as a valid tool for the longitudinal monitoring of patients receiving immunotherapy, although crucially limited by the variable heterogeneity of patients and methodologies [26,27,28]. In this real-world study, in both the all-comers population and the specific treatment subgroups, the patients with higher baseline cfDNA levels showed a significantly shorter median survival, further validating the prognostic value of baseline cfDNA. Consistently, it is well-known that patients presenting with high tumor burden may be associated with high cfDNA levels, thus underlining the prognostic role of cfDNA. Considering the possible risk of bias using only the median value or quartiles, we implemented the X-tile software through training/validation methods to fine-tune the baseline cfDNA threshold values that reliably discriminated the patients who would benefit from first-line systemic treatments. Interestingly, we demonstrated that leveraging the quantitative nature of baseline cfDNA could be a useful clinical tool in the front-line setting of patients with advanced NSCLC, even including mono- and chemo-immunotherapy. Regarding TKIs, we observed a baseline cfDNA cutoff for the PFS significantly higher than that for the OS, probably caused by either higher levels of ctDNA before treatment or the presence of more oncogene-addicted patients who were still alive at the moment of the survival analysis. Furthermore, the smaller sample size as compared to the other treatment subgroups could affect the distribution of estimates around the mean value.

Nonetheless, the cfDNA’s predictive ability in the real-time longitudinal monitoring of NSCLC remains far from clear. We investigated whether a cfDNA response would better correlate with radiologic response and improved survival. Overall, early cfDNA changes seemed to predict the later radiologic response, with a median 20% cfDNA reduction at first restaging significantly associated and consistent over time with the response to front-line treatments. Strikingly, the response ratio of the cfDNA responders (19/27, 70.4%) was 3.7-fold higher than that of the cfDNA non-responders (4/20, 20.0%). However, neither association nor concordance between cfDNA and the radiologic response in the patients receiving CT only was observed. Rather, our findings were statistically and clinically significant in the patients undergoing TKI- or IO-based treatments, dramatically suggesting the predictive role of cfDNA dynamics in these subsets of patients. Moreover, even if a short follow-up must be considered, all the cfDNA responders seemed to benefit the most in terms of the PFS while not showing any trending advantage in the OS, which is probably influenced by later-line treatments. In this regard, cfDNA dynamics could add clinically valuable insights for eventually differentiating pseudoprogression from true progression in patients receiving IO-based treatments [29]. Most importantly, the quantification of cfDNA is easy to obtain and requires minimal processing with well-established isolation procedures, appearing as a reliable and commercially viable biomarker that could be easily implemented in clinical practice, as opposed to other circulating biomarkers, such as circulating tumor cells (CTCs) or tumor mutational burden (TMB), featuring constraining technical challenges that still need harmonization of protocols and standardization of workflows [30]. In this real-world study, the median survival and clinicopathological characteristics seemed to mirror those observed in larger phase III trials [31,32,33]. Consistently, the patients undergoing TKI- or IO-based treatments seemed to experience a significantly improved survival compared to the patients receiving CT only. Moreover, regarding real-life molecular diagnostics on tissue, NGS outperformed single-plex testing in terms of mutation rate detection. Hence, this would suggest a real-world clinical scenario that may be comparable to highly selective randomized clinical trials while additionally including patients with ECOG-PS 2. With phase III trials on immunotherapy and TKIs not enrolling or only including a small proportion of ECOG-PS 2 patients, to date, the negative prognostic outcome of ECOG-PS 2 has been confirmed in only a few retrospective studies evaluating patients on IO-based treatments [34,35,36]. In this first-line prospective cohort, the administration of chemo-immunotherapy in the ECOG-PS 2 patients was significantly associated with a worse PFS and OS compared to single-agent pembrolizumab, further suggesting the use of mono-immunotherapy in PD-L1-high patients with ECOG-PS 2 [34,37]. Of note, together with higher cfDNA levels and liver metastases, in the multivariate analysis, ECOG-PS 2 was associated with worse survival outcomes across all the treatment subgroups. Even if ECOG-PS did not appear to affect cfDNA kinetics, it should be noticed that the ECOG-PS 2 patients presented with significantly higher mean cfDNA levels at baseline.

Limitations of the study included the non-randomized design, the heterogeneity of clinicopathological characteristics (although reflecting a real-world scenario), the small sample size, and the short follow-up together with a single evaluation timepoint, which may have underestimated the final overall results, preventing us from deriving general conclusions.

## 5. Conclusions

In conclusion, our real-world study demonstrates that quantitative changes in cfDNA values correlated with responses to therapy and relapse of the disease in treatment-naïve patients with advanced NSCLC undergoing TKI- and IO-based treatments. The quantification of plasma cfDNA may be a minimally invasive and cost-effective surrogate for improving survival prediction in such patients, retaining only limited clinical utility in patients undergoing CT. However, we do not currently advocate cfDNA analysis as the standard of care in real-time longitudinal treatment monitoring. Nevertheless, this study indicates cfDNA as a reliable biomarker that may help inform the clinical decision-making process along with the design of future larger prospective studies. Larger real-world clinical studies evaluating the predictive role of cfDNA dynamics are warranted before entering the routine clinical application.

## Figures and Tables

**Figure 1 cancers-14-06013-f001:**
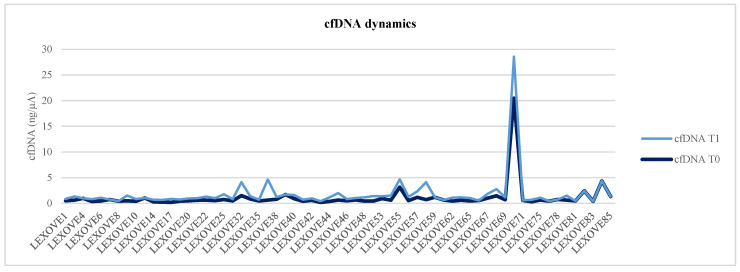
Graphical representation of cfDNA dynamics from baseline (T0) to the radiologic evaluation (T1).

**Figure 2 cancers-14-06013-f002:**
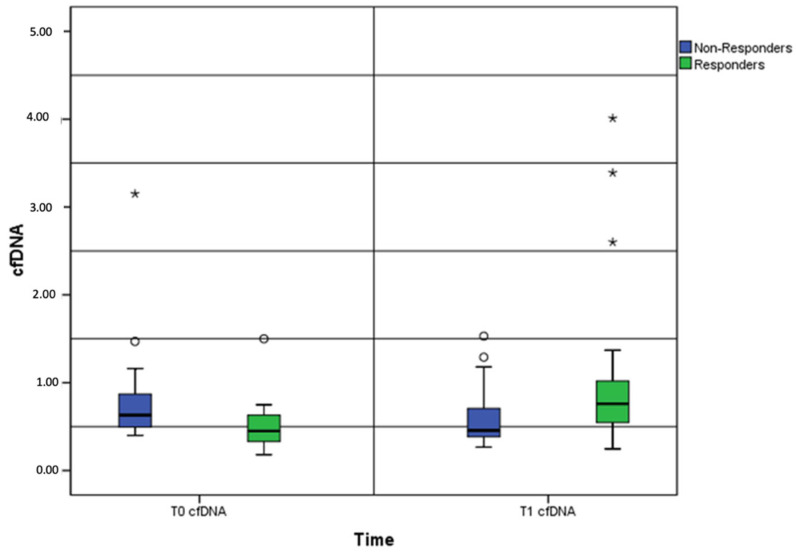
Box-and-whisker plots showing the baseline and post-treatment cfDNA levels between the responders and non-responders. Note: cfDNA, cell-free DNA; *, outliers.

**Figure 3 cancers-14-06013-f003:**
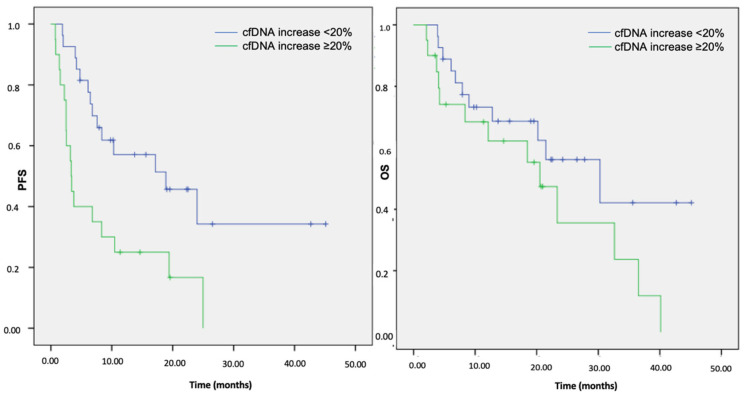
Kaplan–Meier analysis of the PFS and OS according to the cfDNA increase at first restaging in the overall cohort population. Note: cfDNA, cell-free DNA; PFS, progression-free survival; OS, overall survival.

**Table 1 cancers-14-06013-t001:** Baseline patients’ characteristics.

Characteristics	Patients, N (%)
Number of patients	73 (100.0%)
Age, N (%)	
Mean (SD)	67.5 (9.5)
Male	67.3 (10.0)
Female	68 (8.5)
<65 years old (%)	33 (45.2%)
>65 years old (%)	40 (54.8%)
Sex, N (%)	
Male	52 (71.2%)
Female	21 (28.8%)
ECOG-PS, N (%)	
0–1	42 (57.5%)
≥2	31 (42.5%)
Histology, N (%)	
Adenocarcinoma	56 (76.7%)
Squamous cell carcinoma	13 (17.8%)
Others	4 (5.5%)
Smoking history, N (%)	
Never	12 (16.4%)
Former/current	56 (76.7%)
NA	5 (6.9%)
Tumor site, N (%)	
Left	35 (48.0%)
Right	30 (41.0%)
Bilateral	5 (6.9%)
NA	3 (4.1%)
Metastases distribution, N (%)	
Bone	24 (32.8%)
Nodes	62 (84.9%)
CNS	14 (19.2%)
Liver	9 (12.3%)
Adrenal gland	13 (17.8%)
Other	14 (19.2%)
Therapy, N (%)	
TKIs, 19 (26.0%)	EGFR TKIs, 9 (12.3%) ALK TKIs, 5 (6.9%) ROS-1 TKIs, 2 (2.7%) BRAF + MEK TKIs, 3 (4.1%)
IO-based, 28 (38.3%)	Single-agent IO-based treatment, 13 (17.8%) IO-based treatment plus CT, 15 (20.5%)
CT, 26 (35.6%)	Cisplatin–gemcitabine, 2 (2.7%) Carboplatin–gemcitabine, 11 (15.1%) Cisplatin–pemetrexed, 5 (6.8%) Carboplatin–pemetrexed, 8 (11.0%)

**Table 2 cancers-14-06013-t002:** Tissue and plasma molecular diagnostics.

Characteristics			Patients, N (%)	
Number of Patients			73 (100.0%)	
Diagnostic techniques, N (%)				
Tissue, 73 (100.0%)	Real-time PCR,	55 (75.3%)	WT, 44 (80.0%)	
			Mutated, 11 (20.0%)	*EGFR*—8, *BRAF*—3
	NGS	9 (12.3%)	WT, 3 (33.3%)	
			Altered, 6 (66.6%)	*EGFR*—2, *KRAS*—2, *ALK*—1, *MET*—1, *RET*—1, *ROS1*—1
	NA	13 (17.8%)	–	–
Plasma, 10 (13.6%)	Droplet digital PCR	5 (50.0%)	WT, 3 (60.0%)	
			Mutated, 2 (40.0%)	*EGFR*—2
	NGS	5 (50.0%)	WT, 0 (0.0%)	
			Altered, 5 (100.0%)	*EGFR*—2, *BRAF*—1, *KRAS*—1, *EGFR + TP53*—1
Tissue predictive biomarker testing, N (%)				
IHC	PD-L1, 67 (91.7%)	≥50% 1–49% <1% N/A	16 (21.9%) 27 (36.9%) 24 (32.8%) 6 (8.2%)	
	ALK, 56 (76.7%)	Positive Negative N/A	4 (5.4%) 52 (71.2%) 17 (23.2%)	
	ROS1, 46 (63.0%)	Positive Negative N/A	3 (4.1%)—1 confirmed by FISH (1.3%) 43 (58.9%) 27 (37.0%)	
Molecular diagnostics	*EGFR*, 9		p.E746_A750del, 3 p.E746_A750del + p.T790M + p.R175H, TP53, 1 p.E746_A750del + p.C797S, 1 p.L858R, 3 p.L861Q,1	
	*KRAS*, 2		p.G12V, 1 p.G12D, 1	
	*BRAF*, 3		p.V600E, 3	
	*ROS1*, 1		ROS1-CD74, 1	
	*ALK*, 1		EML4-ALK, 1	
	*RET*, 1		KIF5B-RET, 1	
	*MET*, 1		Amplification, 1	
	*NTRK1/2/3,* 0		–	
	*HER-2,* 0		–	
Plasma predictive biomarker testing, N				
Molecular diagnostics	*EGFR*, 2		p.E746_A750del, 1 p.E746_A750del + p.T790M + p.R175H, TP53; 1	
	*BRAF*, 1		p.V600E, 1	
	*KRAS*, 1		p.G12V, 1	

## Data Availability

The datasets used and analyzed in the course of this study are available from the corresponding author upon reasonable request.

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
