# Peer review of "Clinical Potential of Circulating Cell-Free DNA (cfDNA) for Longitudinally Monitoring Clinical Outcomes in the First-Line Setting of Non-Small-Cell Lung Cancer (NSCLC): A Real-World Prospective Study"

_cancers, 2022, doi:10.3390/cancers14236013_

Round 1

Reviewer 1 Report

The authors have studied the cfDNA quantity levels in the plasma of NSCLC patients in relation to the clinical outcomes. The manuscript is interesting as few data are available on cfDNA in NSCLC and because of the great need for non-invasive biomarkers for prognostic and therapeutic follow-up of patients. However, some major and minor revisions to increase the quality of the paper are required.

Minor revisions:

In the Appendix materials, the authors wrote: “FFPE tissue sections with an adequate percentage of neoplastic cells….” The percentage of neoplastic cells within the selected tissue region must be added.

The authors should explain why the difference in the cut-off of cfDNA quantity was found so high between PFS and OS in figure 4,  appendix materials.

The dynamics of cfDNA levels in patients’ follow-up is restricted to a single time point and this, in my opinion, is a great limitation, because more than one single evaluation/patient would be necessary to assess the potency of the cfDNA dynamics at the prognostic level. The authors are invited to evaluate this point and proficiently discuss it.

The baseline and post-treatment cfDNA levels between responders and non-responders (line 240) are more informative by showing them in a  graphic such as a dot plot representation or others similar.

Major revisions:

The authors must show the cfDNA quantity distribution values at baseline between the two age groups of over 65 and under 65. This is because the cfDNA quantity tends to increase with age independently from healthy or pathological status. Thus, it is important to know if the two groups have different baselines. If this would the case, the authors should reconsider the result section accordingly.

Moreover, the authors must show, maybe in a small homogeneous sample of a healthy population, the cfDNA quantity of healthy subjects to compare with that of patients.

Even if the population sample is small, the authors must explore if there are differences in the results shown if only the male population is considered. This is because cfDNA quantity might differ between the two genders. I wonder to know if there is a difference between smokers and non-smokers on cfDNA quantity levels. Finally, the correlation between tumor size and cfDNA quantity must be explored.

The mean age and SD of the male and female groups must be indicated.

The authors must discuss their cfDNA quantity results in terms of ng/ul of plasma with those of literature observed for NSCLC or, if any, for other types of tumors. Moreover, they must deeply discuss the strength of cfDNA quantity levels compared to other known biomarkers underlying the novelty and potentiality.

Author Response

Reviewer 1

In the Appendix materials, the authors wrote: “FFPE tissue sections with an adequate percentage of neoplastic cells….” The percentage of neoplastic cells within the selected tissue region must be added.

R: Thank you for your comment. FFPE tissue sections were considered eligible if the percentage of neoplastic cells was at least 20%. We modified the appendix accordingly.

The authors should explain why the difference in the cut-off of cfDNA quantity was found so high between PFS and OS in figure 4,  appendix materials.

R: Thank you again for your comment. Figure 4 displays the Kaplan–Meier analysis of PFS and OS in patients undergoing TKIs according to the refined threshold based on the X-tyle analysis.  In such oncogene-addicted patients, driven by a molecular target, the quantitative levels of cfDNA are significantly influenced by the high levels of circulating tumor DNA before the treatment start. It must be considered that, while cfDNA clearance in such patients is an early marker of the targeted therapies (PMID: 34250387), at the moment of survival analysis most patients were still alive being under the molecular selective pressure of TKIs. Furthermore, the smaller sample size, as compared to other treatment subgroups, could affect the distribution of estimates around the mean value. Altogether, these aspects could clarify the different cut-off thresholds between PFS and OS. We modified the text accordingly.

The dynamics of cfDNA levels in patients’ follow-up is restricted to a single time point and this, in my opinion, is a great limitation, because more than one single evaluation/patient would be necessary to assess the potency of the cfDNA dynamics at the prognostic level. The authors are invited to evaluate this point and proficiently discuss it.

R: Thank you for your comment. As outlined in our methods, the purpose of the current research was to assess the role of cfDNA dynamics in the minimally invasive monitoring of treatment response according to the first radiological evaluation in previously untreated NSCLC patients. However, as consistently shown by other authors (e.g.: PMID: 29212238), a baseline cfDNA level would be sufficient to evaluate the prognostic significance in terms of both PFS and OS. However, we acknowledged the single-time point follow-up within the limitations. 

The baseline and post-treatment cfDNA levels between responders and non-responders (line 240) are more informative by showing them in a  graphic such as a dot plot representation or others similar.

R: Thank you for your suggestion. We have added a box plot (now Figure 2) to graphically compare baseline and post-treatment levels at glance.

The authors must show the cfDNA quantity distribution values at baseline between the two age groups of over 65 and under 65. This is because the cfDNA quantity tends to increase with age independently from healthy or pathological status. Thus, it is important to know if the two groups have different baselines. If this would the case, the authors should reconsider the result section accordingly.

R: Thank you for your suggestion. We have analyzed the cfDNA dynamics according to age and observed no significant differences between groups. We have added a few lines in the text accordingly.

Moreover, the authors must show, maybe in a small homogeneous sample of a healthy population, the cfDNA quantity of healthy subjects to compare with that of patients.

R: Thank you for your proposal. We did not enroll any healthy controls mostly considering that different meta-analyses including several clinical trials (PMID: 19324991, to name one of all) have already demonstrated that the amount of cfDNA is higher in patients with lung cancer than in healthy individuals with a high level of overall accuracy (PMID: 27597282, PMID: 20004997). These reliable data prompted us to plan a non-randomized pilot trial focusing on the real-life tracking of quantitative levels of cfDNA according to the effect of available standard systemic treatments in consecutively enrolled naïve-treatment patients. Accordingly, other eminent authors have already studied the uncontrolled evaluation of cfDNA dynamics publishing the results in prestigious journals (PMID: 26487589), however not in a perfectly homogeneous first-line real-world setting such as our cohort population. Moreover, as shown by the added subgroup results, no clinical-pathological factors (such as age or gender) seemed to significantly confound the on-treatment results in terms of cfDNA quantitative changes. We have updated the text accordingly.

Even if the population sample is small, the authors must explore if there are differences in the results shown if only the male population is considered. This is because cfDNA quantity might differ between the two genders. I wonder to know if there is a difference between smokers and non-smokers on cfDNA quantity levels. Finally, the correlation between tumor size and cfDNA quantity must be explored.

R: Thanks again for this suggestion. We have verified the absence of significant differences in the subgroups considered (age, sex, and smoking). Furthermore, we have evaluated the correlation between tumor size (in cm) and baseline cfDNA levels. Unsurprisingly, we did not register a positive direct correlation probably due to the contribution of the extrathoracic disease burden which was not the aim of our evaluation. We have updated the text accordingly.

The mean age and SD of the male and female groups must be indicated.

R: Thanks again. Considering the mean (SD), we registered 67.5 (9.5) years in the general population. In addition, the gender subgroup (male and female) were 67.3 (10) and 68 (8.5) years, respectively. We have modified table 1 accordingly.

The authors must discuss their cfDNA quantity results in terms of ng/ul of plasma with those of literature observed for NSCLC or, if any, for other types of tumors.

R: Thank you for the suggestion. We have updated the discussion and references accordingly.

Moreover, they must deeply discuss the strength of cfDNA quantity levels compared to other known biomarkers underlying the novelty and potentiality

R: Thank you for the constructive comment. We have modified the text as follows “Most importantly, the quantification of cfDNA is easy to obtain and requires minimal processing with well-established isolation procedures, rising as a reliable and commercially viable biomarker that could be easily implemented in clinical practice, as opposed to other circulating biomarkers, such as circulating tumor cells (CTCs) or tumor mutational burden (TMB), featured by constraining technical challenges that still need harmonization of protocols and standardization of workflows.[26]”.

Reviewer 2 Report

The present article entitled,"The clinical potential of circulating cell-free DNA (cfDNA) for longitudinally monitoring clinical outcomes in the first line setting of non-small cell lung cancer (NSCLC): a real-world prospective study focusing on ECOG-PS 2 patients" written by Russo et. al., is a well written research work, Although the work has some limitations, the authors have explained them well. Overall, it is a good read.

Author Response

Reviewer 2

The present article entitled,"The clinical potential of circulating cell-free DNA (cfDNA) for longitudinally monitoring clinical outcomes in the first line setting of non-small cell lung cancer (NSCLC): a real-world prospective study focusing on ECOG-PS 2 patients" written by Russo et. al., is a well written research work, Although the work has some limitations, the authors have explained them well. Overall, it is a good read.

R: Thank you for your praise and appreciation.

Reviewer 3 Report

In this study by Gristina et al., the authors examined the prognostic use of circulating free DNA (cfDNA) in monitoring clinical outcomes in NSCLC patients at baseline (and subsequently receiving EGFR TKIs, IO and chemo-therapy) using liquid biopsy. Prospective collections of blood (n=63) and tissue samples were used over a two-year period. Concordance between cfDNA levels and radiological response was examined.

(1) While this study is of interest and important relevance in the lung cancer space in the current era of liquid biopsy and personalized medicine, there are a number of issues that should be more clearly addressed in a revision of the manuscript presented:

(1) The title of the manuscript could be slightly improved to read "The clinical potential of circulating cell-free DNA (cfDNA) for longitudinally monitoring clinical outcomes in the first-line setting....a real-world prospective study". While the study does not focus specifically in ECOG-PS 2 patients, it may be prudent to keep the title more general such as in the above recommedation?

(2) Patient numbers in this prospective cohort are quite low. As such, it is difficult to interpret with confidence the findings presented, albeit, the authors do highlight this as a study limitation in the discussion section.

(3) In the Appendix section, under "FFPE nucleic acid extractions & molecular analysis, the authors state,"...a few ng of DNA and RNA...were tested". The exact amount of DNA/RNA that was used in these analyses should be clearly stated.

In the same Appendix, it reads, "The analytical sensibility..." Is this meant to read "sensitivity?" 

Also, for all numerical values described in the Appendix and main manuscript, these should be written with comma's, as in 30,000 and not 30.000

(4) In the Materials and Methods section relating to ethics, the authors should include the ethics/study approval number from their local ethics committee for this specific study.

(5) On page 3, line 112, correct "IASLC"- not "IASCL"

(6) In the results section of the manuscript, the different cut-offs used for cfDNA are somewhat confusing. In some cases, the median cfDNA value is used, the X-tile analysis in others while furthermore, a cfDNA cut-off using <20% or > or equal to 20% is used, in addition to baseline values of 0.68 ng/ul and 0.61 ng/ul. It is not very clear how these values/cut-offs were determined or where they were derived from.

(7) In Table 1, it would be useful to include the chemotherapy used in these n=26 patients. The table includes this information for TKIs and IO-based treatments, so it would be more consistent to include those for CT also.

(8) Was there significant differences in time between detection of the cfDNA response and that from a clinical radiographic response?

(9) Throughout the manuscript, there are some grammatical errors that need to be corrected such as the words "contrariwise", "in this vein"....

(10) 

Author Response

Reviewer 3

(1) The title of the manuscript could be slightly improved to read "The clinical potential of circulating cell-free DNA (cfDNA) for longitudinally monitoring clinical outcomes in the first-line setting....a real-world prospective study". While the study does not focus specifically in ECOG-PS 2 patients, it may be prudent to keep the title more general such as in the above recommedation?

R: Thank you for your comment.  We have modified the title accordingly.

(2) Patient numbers in this prospective cohort are quite low. As such, it is difficult to interpret with confidence the findings presented, albeit, the authors do highlight this as a study limitation in the discussion section.

R: Thank you for your constructive comment. 

(3) In the Appendix section, under "FFPE nucleic acid extractions & molecular analysis, the authors state,"...a few ng of DNA and RNA...were tested". The exact amount of DNA/RNA that was used in these analyses should be clearly stated.

R: Thank you for your comment.  We used 10 nanograms of both DNA and RNA for performing the NGS analyses. We have modified the text accordingly.

In the same Appendix, it reads, "The analytical sensibility..." Is this meant to read "sensitivity?"

R: Thank you for your comment.  We have corrected the misprint.

Also, for all numerical values described in the Appendix and main manuscript, these should be written with comma's, as in 30,000 and not 30.000

R: Thank you for your clarification. As stated by the MDPI layout style, “the decimal point should always be a dot in numbers” and  “where there are five or more digits to the left of the decimal point”, we used a comma to separate every three digits. Thus, we have thoroughly modified the text.

(4) In the Materials and Methods section relating to ethics, the authors should include the ethics/study approval number from their local ethics committee for this specific study.

R: Thank you for your comment.  We have modified the text accordingly.

(5) On page 3, line 112, correct "IASLC"- not "IASCL"

R: Thank you for your comment.  We have corrected the misprint.

(6) In the results section of the manuscript, the different cut-offs used for cfDNA are somewhat confusing. In some cases, the median cfDNA value is used, the X-tile analysis in others while furthermore, a cfDNA cut-off using <20% or > or equal to 20% is used, in addition to baseline values of 0.68 ng/ul and 0.61 ng/ul. It is not very clear how these values/cut-offs were determined or where they were derived from.

R: Thank you for your clarification. First, we categorized the overall population by the median cfDNA value into low and high groups. Then, to enhance the prognostic accuracy, considering the possible risk of bias using only median or quartiles, we implemented the X-tile software, a tool for the assessment of biological relationships between biomarkers and outcomes, and the discovery of population cut-points based on marker expression (PMID: 15534099). Lastly, we compared baseline and post-treatment cfDNA levels between responders and non-responders, detecting a cfDNA cut-off point of 20% as the median increase from the baseline. We have modified the text accordingly.   

(7) In Table 1, it would be useful to include the chemotherapy used in these n=26 patients. The table includes this information for TKIs and IO-based treatments, so it would be more consistent to include those for CT also.

R: Thank you for the clarification. We have modified the table accordingly.

(8) Was there significant differences in time between detection of the cfDNA response and that from a clinical radiographic response?

R: Thanks for this interesting question. Therefore, we evaluated and compared the time intervals between CT-guided response assessment and liquid biopsy, respectively. The median difference was 8 days, with an unlikely influence on the results shown in the study. We have updated the text accordingly.

(9) Throughout the manuscript, there are some grammatical errors that need to be corrected such as the words "contrariwise", "in this vein"....

R: Thank you for the clarification. We have modified the text accordingly.

Round 2

Reviewer 1 Report

Dear Editor,

The manuscript has been considerably improved and deserves publication in Cancers.

Best regards,

Bruna Scaggiante